# Understanding Retinoblastoma Post-Translational Regulation for the Design of Targeted Cancer Therapies

**DOI:** 10.3390/cancers14051265

**Published:** 2022-02-28

**Authors:** Radoslav Janostiak, Ariadna Torres-Sanchez, Francesc Posas, Eulàlia de Nadal

**Affiliations:** 1Institute for Research in Biomedicine (IRB Barcelona), The Barcelona Institute of Science and Technology, Baldiri Reixac, 10, 08028 Barcelona, Spain; radoslav.janostiak@irbbarcelona.org (R.J.); ariadna.torres@irbbarcelona.org (A.T.-S.); 2Department of Medicine and Life Sciences (MELIS), Universitat Pompeu Fabra (UPF), 08003 Barcelona, Spain

**Keywords:** retinoblastoma, cell cycle, cancer therapies

## Abstract

**Simple Summary:**

Rb1 is a regulator of cell cycle progression and genomic stability. This review focuses on post-translational modifications, their effect on Rb1 interactors, and their role in intracellular signaling in the context of cancer development. Finally, we highlight potential approaches to harness these post-translational modifications to design novel effective anticancer therapies.

**Abstract:**

The retinoblastoma protein (Rb1) is a prototypical tumor suppressor protein whose role was described more than 40 years ago. Together with p107 (also known as RBL1) and p130 (also known as RBL2), the Rb1 belongs to a family of structurally and functionally similar proteins that inhibits cell cycle progression. Given the central role of Rb1 in regulating proliferation, its expression or function is altered in most types of cancer. One of the mechanisms underlying Rb-mediated cell cycle inhibition is the binding and repression of E2F transcription factors, and these processes are dependent on Rb1 phosphorylation status. However, recent work shows that Rb1 is a convergent point of many pathways and thus the regulation of its function through post-translational modifications is more complex than initially expected. Moreover, depending on the context, downstream signaling can be both E2F-dependent and -independent. This review seeks to summarize the most recent research on Rb1 function and regulation and discuss potential avenues for the design of novel cancer therapies.

## 1. Introduction

The discovery of the Retinoblastoma-associated protein 1 (Rb1) in the late 1970s was an important milestone in understanding cancer development [1,2,3]. Until then, cancer was believed to arise because of dominantly activated oncogenes [4]. However, studies of hereditary retinoblastoma tumors brought about the discovery of Rb1 as a prototype tumor suppressor gene whose homozygous inactivation leads to the development of tumors [5,6,7]. The *Rb1* locus is located on the q arm of chromosome 13, and the presence of genetic rearrangements in this region in retinoblastoma tumors was one of the crucial pieces of evidence for the identification of Rb1 as a key player in the development of such tumors [6,7].

The product of *Rb1* comprises 928 amino acids, has a molecular weight of 106 kDa, and belongs to a family of structurally related molecules called pocket domain proteins. The family consists of Rb1 and its two paralogs Rbl1 (p107) and Rbl2 (p130), which share similar domain organization, although several differences make Rbl1 and Rbl2 more related to each other than to Rb1 [8,9].

Rb1 protein has several domains that are crucial for its function and interactions. In the past, the central pocket domain (residues 379—792) and C-terminal domain (residues 792—926) were considered the most important parts for Rb1 function [10], namely arrest of cell cycle progression through inhibition of E2F1 function [11]. In addition to its canonical function, Rb1 plays a key role in regulating other cellular processes such as differentiation, maintenance of genome stability, and immune evasion [12,13,14]. Rb1 function is deregulated in most types of human cancer through genetic alteration or, more often, through alteration of upstream pathway components that inactivate its function through post-translational modifications [15,16].

In this review, we describe the canonical and non-canonical functions of Rb1, its alteration in human cancers, and the post-translational modifications that are crucial for its regulation. Finally, we discuss how Rb1 post-translation modifications could be harnessed to design novel anticancer therapies.

## 2. Rb1 Function and Downstream Signaling

To date, more than 300 Rb1 interactors have been identified, thus indicating that this protein is involved in multiple signaling complexes [17]. Although the most studied signaling event is the cell cycle-dependent binding of Rb1 to E2F transcriptional factors (E2F-dependent signaling), Rb1 regulates other signaling pathways that can be roughly classified as E2F-independent.

### 2.1. E2F-Dependent Signaling

The key event for the regulation of E2F1-dependent downstream signaling is the binding of Rb1 to E2F1. This interaction and its consequences have been studied extensively and the mechanisms involved are well characterized.

Initial binding analyses and structural experiments determined two structural features that delineate the binding of Rb1 to E2F1, namely the central pocket and C-terminal domains [10,18,19] (Figure 1A). The central pocket domain of Rb1 is responsible for binding to E2F1. The relevance of this domain for this binding and for regulating E2F1-mediated transcription was shown by in vitro and in vivo studies, as well as by crystallography. The central pocket domain of Rb1 (residues 379—792) consists of two cyclin folds (Boxes A and B) connected by a flexible linker [10]. This region binds to an 18 amino acid-long peptide within the transactivation domain of E2F1 (residues 409—426) [10]. This interaction is crucial for binding, as well as for inhibiting E2F1 transcriptional activity in vitro; however, it is insufficient to fully inhibit E2F1-mediated transcription in vivo, thereby suggesting further interaction requirements [20,21,22]. This additional structural and functional interaction is provided by the C-terminal domain of Rb1, which does not adopt a particular structure by itself. Instead, two regions in this domain (residues 786—800 and 829—846) bind to a heterodimeric structure formed by the E2F1 coiled-coil marked box (CCMB) domain (residues 200—301) and to the CCMB domain (residue 199—350) of the transcription factor DP1 (TFDP1), to form a stable complex [23]. The binding of Rb1 to E2F1 then results in inhibition of E2F1-mediated transcription of cell cycle progression genes, such as *Cyclin A*, *Cyclin E* and *cdc25*, which ultimately leads to cell cycle arrest and inhibition of cell proliferation [11,24,25]).

Structure and interaction surfaces are regulated by phosphorylation by CKDs. There are two main regions on Rb1 that are phosphorylated by CDKs—central pocket domain and C-terminal domain. Phosphorylation of residues within these regions (T356, T373, S788, S795, S807, S811, S821, S826) leads to disruption of Rb1 with E2F1 interaction and transcriptional activation of cell cycle regulating genes (More detailed description of these phosphorylations is in following sections).

Rb1-mediated inhibition of E2F transcriptional activity is mediated by two distinct molecular mechanisms. First, Rb1 binding to the transactivation domain of E2F blocks the recruitment of general transcription factors and histone acetylases that are necessary for transcription initiation [26,27]. Second, Rb1-bound E2F recruits numerous histone-modifying and chromatin-remodeling factors to E2F-target promoters, thus altering chromatin structure and configuration. It was shown that these factors contain the so-called LXCXE motif, which associates with a structure known as the “LXCXE- binding cleft” within the Rb1 central pocket domain [19,28]. Through this domain, Rb1 binds and recruits several members of histone deacetylases (HDACs) and histone methylases. To date, the interaction of Rb1 and four members of the HDAC family (HDAC1, 2, 3, 5) has been described [28,29,30,31] and evidence points to the interaction of Rb1 and HDACs being crucial for transcriptional silencing of E2F1-target promoters.

After the recruitment of HDACs to E2F-target promoters, the deacetylases act on the surrounding histone tail residues, thus repressing E2F-mediated transcription. Moreover, this repression is dependent on Rb1 and also on HDAC activity, as trichostatin A treatment (HDACi) impairs this inhibition [29,30]. Recently, the interaction between HDAC5 and Rb1 was reported to be crucial for the transcriptional repression of pro-oncogenic genes. Furthermore, deletion of HDAC5 results in the impairment of Rb1-mediated silencing of cell cycle genes and conferred resistance to CDK4/6 inhibitors [31]. Another histone modification regulated by Rb1 is methylation. Rb1 binds to the lysine methyl transferase SUV39H [32]. Rb1-mediated recruitment of SUV39H to the E2F1-target promoters leads to trimethylation of H3K9, creating a binding site for heterochromatin protein 1 (HP1), which triggers transcription silencing [32,33]. In addition to histone-modifying enzymes, Rb1 also recruits helicases Brm and Brg1, which are indispensable for the functionality of the SWI/SNF chromatin remodeling complex [34,35,36]. As a result of this recruitment, chromatin is packed more tightly, which adds another layer of transcriptional repression of E2F1-targeted genes and causes inhibition of cell cycle progression [37]. Finally, repression of E2F1-stimulated transcription can be mediated by Rb1-dependent recruitment of DNA methyltransferase 1 (DNMT1). DNMT1 recruitment is dependent on the amino-terminal part of DNMT1 (amino acids 416-913) and the intact pocket domain of Rb1 [38] (Figure 1B).

An interesting expansion of the Rb1 function is the regulation of silencing of repetitive DNA sequences, such as endogenous retroviruses or LINE-1 elements. These elements bind E2F1, which results in cell cycle-independent recruitment of Rb1. Epigenetic silencing of these elements is dependent on H3K27 trimethylation and it is markedly reduced in cells where Rb1 recruitment is impaired. The underlying mechanism is Rb1-dependent recruitment of EZH2 to the repetitive elements. Moreover, E2F1 binding-deficient Rb1 was still able to associate with EZH2, however, there was no deposition of methylation to the repetitive sequences, which confirms that maintenance of epigenetic silencing of the repetitive sequence is dependent on the formation of the E2F1–Rb1–EZH2 tricomplex [39].

The interaction of Rb1 with E2F is regulated by either post-translational modification of Rb1, which is discussed in the next section, or by binding with certain cellular and viral proteins. Among the latter, viral proteins, such as adenovirus (Ad) E1A, human papillomavirus (HPV) E7, and simian virus 40 (SV40) large T antigen (LTa), have been characterized in most detail [40]. Although diverse, these small viral proteins share a common LxCxE sequence that binds to the pocket domain of Rb [41,42,43,44]. Evidence indicates two general mechanisms by which these small viral proteins relieve the inhibition of Rb1 transcription. Initially, it was proposed that these viral proteins competed for the binding surface that interacts with E2F, thus interfering with the formation of the Rb1–E2F inhibitory complex [40]. However, later structural and functional studies revealed that another plausible mechanism is that these viral proteins interfere with the binding of Rb1 with HDACs through competition for the same docking site, namely the LxCxE binding cleft [29,45]. This interference results in the inactivation of the Rb1 function and failure to repress E2F transcription because of high histone acetylation and an open, permissive state of chromatin [46,47].

In addition to E2F1, Rb1 can bind other members of the E2F family, thereby adding further complexity to the role of this protein. E2F2, E2F3, E2F4 and E2F5 members contain Rb1-binding domains similar to E2F1 and interact with Rb1 [48,49]. The interaction of Rb1 with E2F2 and E2F3 results in blockage of their pro-proliferative function, which is consistent with their established role as transcriptional activators [48,50]. However, E2F4 and E2F5 are considered transcriptional repressors and therefore the function of their interaction with Rb1 is different. The Rb1–E2F4/5 interaction occurs mainly in the G0 phase, where the two proteins form an inhibitory complex on the E2F1/2/3-promotors to block E2F1/2/3-stimulated transcription and maintain cell quiescence [51]. Indeed, it was shown that the main function of E2F4/5 is to regulate the development and terminal differentiation of various cell types [52,53,54].

### 2.2. E2F-Independent Signaling

The mechanism by which Rb1 regulates E2F-mediated transcription has been extensively studied; however, Rb1 binds to multiple other cellular proteins and alters their function independently of E2F, thereby affecting cell cycle progression, differentiation and transcription.

The E2F-independent role of Rb1 is well established in the context of lineage commitment and terminal differentiation of different cell types. Rb1 interacts with several transcription factors, such as MyoD, Runx2, C/EBP, NF-IL6, GATA1 and Pax8, and acts as a transcriptional co-activator. This interaction subsequently leads to stimulation of lineage-specific gene transcription and progenitor differentiation into mature tissue-specific cells, such as adipocytes, myocytes, hematopoietic cells and osteocytes [12,13,14,55,56,57]. The interaction between these transcription factors and Rb1 is mediated by the LxCxE sequence and Rb1 pocket domain, and mutation of this domain or overexpression of pocket domain-binding small viral proteins (E7, E1A) disrupt this interaction. The disruption of the interaction between Rb1 and these lineage-specific transcription factors results in impairment of differentiation, re-entry into the proliferative mode, and cancer development [12,14,55]. A complementary mechanism by which Rb1 stimulates terminal differentiation is by sequestering transcription factors that inhibit differentiation, such as EID-1 and ID2 (inhibitors of myocytes and osteocytes differentiation, respectively) [58,59] (Figure 1B).

Another important transcription factor bound by Rb1 is p65; however, there is no definitive consensus on the nature of this interaction. An initial study conducted on osteosarcoma cells showed that Rb1 binds and stimulates transcriptional activity of p65, which leads to an increased expression of cyclin D1 [60]. On the other hand, a more recent study revealed that Rb1 binds p65 in response to CDK-mediated phosphorylation and inhibits its transcriptional activity specifically towards PD-L1, which confers cancer immune evasion. Moreover, the authors showed that this interaction is mediated through the Rb N-terminal domain and the CCMB domain of p65 [61]. Thus, as observed for E2F factors, in some cases the binding of Rb1 induces transcription while in anothers it represses transcription, thereby suggesting a gene context-dependent effect of Rb1.

Another process directly associated with transcription control is an epigenetic modification of chromatin. As described above, Rb1 can recruit chromatin-modifying enzymes to E2F-target gene promoters and thus alter E2F-stimulated transcription. However, some reports indicate that Rb1 facilitates the function of several histone modifiers also in an E2F-independent manner. One such enzyme is histone demethylase KDM5a (RBP2), which demethylates tri- and di-methylated H3K4 [60]. The interaction of Rb1 and KDM5a is mediated by the classical LxCxE—pocket domain-binding interface. This interaction is important for myocyte specification, thereby supporting the crucial role of Rb1 in differentiation [62]. In addition to local facilitation of chromatin reorganization, Rb1 plays a key role in global heterochromatin maintenance. In this regard, fibroblasts deficient in Rb1 expression or expressing Rb1 with a deletion in the LxCxE binding cleft show highly decondensed chromatin, even in the centromeric regions [63]. More specifically, impaired heterochromatin maintenance of these fibroblasts was associated with a decrease in H4K20 methylation. This histone mark is deposited by histone methyltransferase SUV420H, which directly interacts with Rb, thus heterochromatin dissolution is most probably a result of disruption of the Rb1-SUV420H interaction [64].

In addition to the CDK–Rb1–E2F axis, Rb1 regulates cell cycle progression via several E2F-independent mechanisms. One such mechanism is the interaction with ubiquitin ligase complexes SCF and APC/C, which facilitate the transition between phases of the cell cycle [65,66]. Rb1 binds to both types of complexes and regulates the stability of the CDK inhibitor p27Kip1 via two distinct mechanisms. The SCF (SKP1-Cullin-F box) complex is a crucial regulator of the G1 phase transition and, when active, degrades cell cycle inhibitors, such as p21, p27 and p57 [66]. SKP2 is a crucial component of this complex, and it was shown to interact with the C-terminal part of Rb1. The sequestration of SKP2 by Rb1 leads to the release of p27 from the SCF^Skp2^ ubiquitination complex and its stabilization. Increased levels of p27 subsequently result in the inhibition of CDK activity and cell cycle arrest [67]. The APC/C complex is another multisubunit ubiquitin ligase that is active from the onset of mitosis through to the late G1 phase [65]. Rb1 interacts with the Cdh1 subunit of the APC/C complex simultaneously with SKP2. This bridging interaction then facilitates the ubiquitination of SKP2 by APC/C, which in turn leads to the stabilization of p27. The outlined series of events results in CDK inhibition and exit from the cell cycle [68,69] (Figure 1B).

## 3. The Role of Rb1 in Cancer Development and Progression

Since the description of Rb1 as the first tumor suppressor protein, its genetic and epigenetic alteration and function have been extensively studied and characterized. As mentioned, its canonical function through its binding to E2F and transcriptional repression leads to the arrest of cell cycle progression. This inhibitory regulation is impaired in most types of cancer either through alteration of Rb1 expression (deletion, mutation, epigenetic silencing) or more often through functional inactivation by CDK-stimulated hyperphosphorylation. In addition to this canonical function, Rb1 is also important for the maintenance of genomic and chromosomal stability. These roles of Rb1 are addressed in the following sections.

### 3.1. Genetic and Epigenetic Alteration of the Rb1 Locus

Alteration in the gene locus coding for Rb1 protein is considered the first evidence showing the crucial role of tumor suppressor genes in cancer development. It was established very early that the homozygous deletion of Rb1 is the major genetic alteration driving the development of childhood retinoblastoma due to loss of Rb1 function as a cell cycle regulator [6,70]. Afterward, genetic alteration of the *Rb1* locus was associated with the development of various types of cancer, including sarcoma, lung and ovarian cancer [71]. Analysis of mutation types and frequencies in patient-derived cancer samples deposited in the cBio Cancer Genomics Portal (cbioportal.org; accessed on 8 February 2022) database revealed that *Rb1* is altered (point mutations, deletion, amplification) in 7% of all samples. However, mutational frequencies vary between tumor types, with bladder cancer and sarcoma samples having the highest proportion of Rb1 mutations (20–25%), followed by prostate cancer, melanoma, and liver and brain cancer (10–15%). Most of these mutations (88%) are classified as driver mutations, which result in the absence of functional Rb1 protein (*Rb1* locus deletion—41%; Rb1 truncating mutation present—33%, Rb1 splice variant mutation present—12%). These frequencies are consistent with Rb1 being a negative regulator of the cell cycle. Moreover, 16% of the alterations are missense mutations (Figure 2). These are distributed throughout the whole gene with no apparent recurrent mutations in either of the residues or domains [15,16]. The significance of these point mutations is unclear, and only a few of them have been studied. For example, mutations Leu607Ile, Arg698Trp, and Arg621Cys identified in breast cancer patients were shown to have impaired pro-apoptotic function of Rb1 and they stimulated resistance to 5-FU/mitomycin or doxorubicin [72]. Interestingly, different Rb1 mutations were identified as a result of the treatment of breast cancer patients with CDK inhibitors and were considered to confer therapeutic resistance [73,74]. Moreover, the association of these mutations with the clinical outcome of patients with cancer is not well established, thus further complicating the role of Rb1 in cancer. However, mutations in upstream regulators of Rb1 (cyclins, CDKs or CDKis) are more prevalent and are associated with a strong and clear clinical outcome, namely worse prognosis and shorter survival [75,76,77,78].

Another common mechanism for the downregulation of Rb1 expression is hypermethylation of its promoter region. This methylation was first described in retinoblastoma patients, and in vitro experiments confirmed that *Rb1* contains a CpG island, which, when methylated, strongly reduces Rb1 expression [79]. Mechanistically, methylation abolishes the binding of activating transcription factor (ATF)-like factor and retinoblastoma-binding factor 1 (RBF1) to their cognate sequences [80]. Subsequent methylation-specific screening efforts revealed Rb promoter hypermethylation in 20–25% of brain tumors and 11% of retinoblastomas [81,82,83].

### 3.2. Canonical Tumor Suppressor Role of Rb1

However, the phosphorylation of Rb1 by CDKs is the most common event for Rb1 inactivation. Phosphorylation by CDKs serves to dissociate the binding of Rb1 to E2F (Figure 1). Correspondingly, oncogenic activation of CDK4/6 either by activating mutations in *CDK4/6*, gene amplification of positive regulators such as CCND1, or deletion of CDK4/6 inhibitors such as p16, results in functional inactivation of Rb1 protein and the expression of genes that are important for mitosis and cell cycle progression [84,85,86,87,88] (Figure 2). In line with this evidence, clinically approved CDK4/6 inhibitors (e.g., Palbociclib, Ribociclib, Abemaciclib) significantly impair tumor cell proliferation, especially in ER+ breast cancer [89,90]. This inhibitory effect is almost exclusively dependent on the presence of Rb1 protein [91,92]. The clinical efficacy of CDK4/6 inhibitors is highly positively correlated with the expression of Rb1, and the loss of functional Rb1 protein through locus deletion or point mutations is a common mechanism of acquired resistance to CDK4/6 inhibitors [74,93]. However, other signaling alterations can lead to CDK4/6 inhibitor resistance. Among these, the upregulation of Cyclin D expression or amplification of Cyclin E is clinically relevant and leads to the inactivation of Rb1 [75,94,95,96,97]. This absolute requirement of Rb1 inactivation for successful cell cycle progression offers a therapeutic opportunity to develop compounds to directly stimulate the Rb1 function by increasing its binding to its partners, therefore maintaining a more sustainable and stronger cell cycle exit.

Another mechanism by which Rb1 protects cells from malignant transformation is through the maintenance of proficient DNA damage repair and chromosomal stability. Several studies report the involvement of Rb1 protein in DNA double-strand break (DSB) repair by all three major DSB repair pathways—homologous recombination (HR), non-homologous end joining (NHEJ), and microhomology-mediated end joining (MMEJ). Rb1 localizes to the sites of DNA damage in an E2F1- and ATM (serine-threonine kinase ataxia-telangiectasia mutated)-dependent manner. Mechanistically, ATM-phosphorylates E2F1 at S29, which leads to its recruitment to p-γH2AX foci. Rb1 is localized to these foci via association with E2F1, and Brg1 DNA helicase is subsequently recruited in an Rb1-dependent fashion [98]. Recruitment of Brg1 results in nucleosome repositioning, allowing proper DSB repair to take place [99]. Rb1 also stimulates DSB repair by promoting end resection and the HR and NHEJ repair pathways. Rb1 colocalizes with CtIP (DNA endonuclease RBBP8) at the sites of DNA damage and stimulates CtIP phosphorylation on T847, CtIP loci formation, and RPA (Replication Protein A) loading [100]. These events subsequently result in DNA break repair [101]. Moreover, by interacting with key components of NHEJ (XRCC5 and XRCC6), Rb1 is directly involved in the stimulation of NHEJ repair [102]. Taken together, these studies show that Rb1-deficient cells are hypersensitive to DNA damage agents, and this feature may be useful for designing therapeutic strategies based on synthetic lethality.

In addition to its direct role in the DSB repair, Rb1 also facilitates genome integrity through stabilizing heterochromatin. As described earlier, Rb1 interacts with and recruits several epigenetic factors that alter chromatin structure. It is now well established that Rb1 plays a crucial role in stabilizing heterochromatin at centromeric and telomeric regions through maintenance of sufficient levels of H4K20 methylation. Rb1-deficient mouse fibroblasts or fibroblasts expressing Rb1 mutated in the LxCxE binding cleft show decondensed chromatin and decreased H4K20 levels specifically at telomeres and pericentric chromatin [63,64]. This effect is independent of E2F1 and is most probably facilitated by the recruitment of Suv420h1/2, which is a known methyltransferase for H4K20. Moreover, this aberrant H4K20 methylation is associated with impaired HP1 (Heterochromatin Protein 1), which is crucial for heterochromatin stabilization at telomeres and pericentric regions [103]. In addition to stabilizing heterochromatin, several reports established a clear connection between Rb1 and Cohesin II recruitment. *Rb1* dosage is strongly associated with the protection of cells from chromosomal alterations and aneuploidy. Full Rb1 deletion leads to a high rate of chromosomal instability, and even deletion of a single copy has a profound effect. Mechanistically, Rb1 recruits Cohesin II to the centromeric regions and thus promotes faithful segregation of sister chromatids [104,105]. In conclusion, besides its well-established role in regulating G1/S progression, Rb1 is relevant for the maintenance of chromosomal stability throughout the whole cell cycle, including mitosis. Abolishing Rb1 functionality thus leads to chromosomal instability (CIN), which further fuels cancer progression [106,107,108].

### 3.3. Non-Canonical Function of Rb1

Finally, yet importantly, some functions of Rb1 can be appreciated only in the context of the cancer microenvironment and interaction with immune cells. Several reports demonstrate the involvement of the E2F1/Rb1 axis in the regulation of antitumor immune response. Most of these focused on the outcome of systemic CDK4/6 inhibition, which was evaluated mainly as the effect on immune cell proliferation and recruitment [109,110,111,112]. However, the role of Rb1 in modulating intrinsic tumor immunogenicity has not been thoroughly addressed. As described above, Rb1 inactivation triggers increased genomic instability [104,113,114], which can subsequently lead to the generation of proteins with altered sequences that can serve as tumor-associated or tumor-specific antigens, which would be presented on the surface of cancer cells [115]. These neoantigens could then stimulate immune cell recruitment and activation, thus leading to cytotoxicity-mediated cell death [116]. Indeed, tumors with a higher mutagenic burden produce greater amounts of tumor neoantigens [117,118,119]. Such immunologically hot tumors are characterized by a higher number of infiltrating lymphocytes and better clinical efficacy of therapy targeting immune checkpoint inhibitors, such as PD/PD-L1 or CTLA4 [120,121,122].

Similarly, and as mentioned above, it was shown that Rb1 participates in regulating PD-L1 expression; however, this function is paradoxical. A recent study revealed that CDK4/6-mediated phosphorylation of Rb1 on residues S249 and T252 increases its affinity to p65 and inhibits its transcriptional activity. Since one of the p65 targets is PD-L1, this inhibition leads to decreased PD-L1 expression and subsequent impairment of immune evasion [61].

Moreover, Rb1 binds to EZH2 and alters the epigenetic modification status of certain sequences [39]. Such inhibitory complexes can be formed on the promoters of certain surface ligands that are important for immune cell recruitment. Indeed, there is substantial evidence that epigenetic modifiers such as EZH2 play a crucial role in regulating this process (see review [123]).

## 4. Upstream Regulation and Post-Translational Modifications

Given that Rb1 was described as a prototypical tumor suppressor more than 40 years ago, there is a huge amount of evidence regarding its post-translational modifications. According to phosphositeplus.org, there are 105 modified residues in Rb1. Phosphorylation (61 residues) and ubiquitination (29 residues) are the most common modifications of this protein, while others (acetylation, methylation) have received less attention. The following chapter describes these post-translational modifications, along with the upstream signaling and downstream significance (Figure 3, Table 1).

### 4.1. Phosphorylation

The prototypical and most widely studied post-translational modification of Rb1 is phosphorylation. It was the first modification identified to be crucially important for Rb1 function as a tumor suppressor [160]. To date, 61 residues (serine, threonine, tyrosine) have been shown to be phosphorylated by several kinases, such as Cyclin-dependent Kinases (CDKs), p38 and Aurora B kinase (AurKB) [94,125,129,161], 19225156. A textbook example of phosphorylation-related modulation of Rb1 function is the phosphorylation of serine and threonine residues in the C-terminal domain of the protein by CDKs. These phosphorylations lead to the inactivation of Rb1, disassociation of the E2F transcription factor, and the induction of transcription of genes regulating cell cycle entry and progression [161]. Proteomic and mutational analysis of the Rb1 residues revealed 16 putative CDK-targeted phosphorylation residues, of which 15 are located on the surface and are thus accessible to phosphorylation [162,163]. Although all of these sites can be phosphorylated by individual CDKs, the surrounding sequence plays an important role in specificity and selectivity. Mechanistically, it is generally accepted that cells in the G0 phase of the cell cycle do not contain phosphorylated Rb1 on any of the residues [164,165], although a degree of basal phosphorylation of these sites is constant, as shown by mass spectrometry analyses. However, as cells enter the cell cycle, the phosphorylation pattern becomes more complex, and it is even possible to talk about an Rb1 phosphorylation code [163]. Historically, it was shown that Rb1 is initially mono-phosphorylated at S249, T252, T356, S608, S788, S807, S811 and S826 by CDK4/6-Cyclin D throughout the G1 phase, thus priming it for further phosphorylation again by the same complex or by CDK2-Cyclin E at T5, T373 and S795 [166]. Moreover, it was also reported that phosphorylation of Rb1 on residues S807/811 by CDK3 is necessary for the G0–G1 transition [167]. Later, at the end of the G1 phase, the CDK2-Cyclin E complex phosphorylates S612 and T821, which results in fully hyperphosphorylated Rb1, which then dissociates from E2F, thus allowing progression to the S phase [166].

Since E2F1 was considered the most important Rb1 interaction partner, the effect of phosphorylation of these residues on the Rb1–E2F1 interaction was studied mainly with a focus on the interaction with E2F1, leaving the possibility that some of the phosphorylations may be more relevant for alternative binding partners. Initially, it was reported that phosphorylation of S788 and S795 directly inhibits the binding of Rb1 to E2F1, whereas phosphorylation of T821 and T826 facilitates the closing of the Rb1 structure as a result of increased binding of its C-terminal and central pocket domain [23]. Furthermore, phosphorylation of T373 causes a structural change that increases intramolecular interaction between the Rb N-terminal and the pocket domain. Finally, phosphorylation of S608 results in the burying of the loop with CDK target sites within the Rb1 pocket domain, thus impairing hyperphosphorylation [168]. These findings were further corroborated by evidence that most CDK target sites on Rb1 are phosphorylated in the M-phase of the cell cycle [157,169]. However, this simplistic view of phosphorylation dynamics was challenged by a comprehensive analysis of the phosphorylation status of individual residues and their function. Narasimha and colleagues showed that Rb1 is exclusively monophosphorylated throughout the G1 phase and that Cyclin D–CDK4/6 complexes mediate this phosphorylation. Moreover, the study revealed that monophosphorylation can occur on any of the CDK-target sites, but always as a single event. At the restriction point, CDK2/Cyclin E complexes then hyperphosphorylate Rb1, thus allowing progression to S-phase [162]. These results were verified by a later study that showed that even single phosphorylation of Rb1 by CDK6 greatly reduces the capacity of Rb1 to further interact with Cyclin D/CDK6 complexes [163]. Interestingly, after DNA damage, Rb1 is monophosphorylated again by the action of the Cyclin D/CDK4/6 complex [162].

Since Rb1 is a major regulator of the cell cycle, it integrates signals from stress response pathways such as DNA damage and environmental stresses. It was shown that S612 is phosphorylated by Chk1/2 in response to DNA damage and that this phosphorylation increases its affinity towards E2F1 [133]. Similar to phosphorylation at S608, S612 phosphorylation could involve shielding of the loop containing hyperphosphorylation sites that increase the affinity to E2F1 [133]. However, the phosphorylation of these sites is highly context-specific because phosphorylation of S608/S612 together with T356/T373 leads to the disruption of Rb1 and E2F1 binding [128], again suggesting a complex code for Rb1 phosphorylation and regulation of its activity. An important function of Rb1 phosphorylation in sensing DNA damage is underlined by evidence showing that functional inactivation of Rb1 by deletion or mutation leads to increased sensitivity to DNA-damaging agents [170,171,172].

Furthermore, Rb1 is a direct target of the stress-activated kinase p38α, which integrates stress signaling beyond DNA damage. p38α is a master regulator of stress signaling pathways [173] and, indeed, several studies report p38α-mediated Rb1 phosphorylation. An initial study indicated that p38α stimulates phosphorylation of Rb1 at S807/811 after mitogenic stimuli such as serum addition. However, this phosphorylation seems to be indirect through stimulation of Cyclin D [174]. Stronger evidence showed that p38α directly phosphorylates five residues on Rb1 (S249, T252, S576, S838, T841). Of note, Rb1 is phosphorylated at S249/T252 by p38α in response to environmental stress, such as osmostress, oxidative stress, and DNA damage [125,175]. Additionally, induction of TGFβRII in metastatic prostate cancer cells mediates RB S249/T252 phosphorylation by p38 which prevents bone metastasis [176]. A study analyzing the global change of phosphorylation after NaCl-induced stress showed increased phosphorylation of these residues [177]. In contrast to the phosphorylation by CDKs, S249/T252 phosphorylation increases interaction between Rb1 and E2F1, thus causing cell cycle arrest. An Rb1 mutant that cannot be phosphorylated by p38α on those sites is deficient in cell cycle delay upon stress, and cells show a clear reduction in viability. Of note, the increased affinity of E2F-Rb1 due to S249/T252 phosphorylation is also observed in the presence of high CDK activity. Thus, this phosphorylation can override the phosphorylation in central and pocket domains of Rb1 by CDKs that favor Rb1/E2F1 disassociation, thus representing a higher regulatory level to stop the cell cycle in unfavorable environmental conditions [125]. Of note, cells expressing a mutant carrying a phosphomimetic mutation on those sites show reduced tumorigenesis in vitro and in vivo [125].

In addition to p38α, other reports showed that S249/T252 can be also phosphorylated by CDKs, which results in altered binding with other Rb1 interaction partners such as p65 and HDAC5 [31,61]. This particular case further highlights the complexity of Rb1 regulation, which appears to be context-dependent and to be influenced by the concomitant phosphorylation of other residues of Rb1, affecting the overall outcome of single phosphorylation in each case. Indeed, a recent study analyzing the role of monophosphorylated Rb1 showed that distinct monophosphorylation leads to different signaling outcomes [163].

Another well-described p38α-directed Rb1 phosphorylation site is S567. This phosphorylation is triggered by DNA damage and it results in the disassociation of Rb1 from E2F1, increased interaction with HDM2 ubiquitin ligase, Rb1 poly-ubiquitination and subsequent targeting to proteasomal degradation [129,130]. This signaling requires additional regulatory circuits that block the cell cycle-promoting function of released E2F1 but allow the stimulation of apoptosis [20,178]. This observation highlights that there might be Rb1 fractions with different phosphorylation patterns depending on the promoter regions (cell cycle vs. apoptotic) in which these particular Rb1/E2F1 complexes are located. The most recently identified p38α-targeted phosphorylation sites are S838 and S841 [144]. This particular phosphorylation was studied in immune cells in response to T-cell receptor activation. p38α-mediated Rb1 phosphorylation of these sites results in disruption of Rb1 and Condensin II association and subsequent chromatin relaxation. However, the overall significance of this phosphorylation is not clear, since it was also identified as a target of Aurora kinase in cell types of non-immune origin [150].

Of note, the isoform p38γ cooperates with CDKs, regulating entry into the cell cycle. In mouse hepatocytes, p38γ induces proliferation after partial hepatectomy by promoting the phosphorylation of Rb1 at known CDK target residues. Lack of p38γ or treatment with the p38γ inhibitor protects against the chemically induced formation of liver tumors [179].

Rb1 is directly phosphorylated by the mitotic kinase Aurora B at S780. This phosphorylation is important for mediating the role of Aurora B in regulating the postmitotic duplication checkpoint, thus protecting the cells from aneuploidy [138]. These results are well aligned with the Rb1 function in protecting genome integrity discussed above. Interestingly, the deletion of Rb1 is synthetically lethal with deletion/inhibition of both Aurora A and B kinases [180,181,182].

In contrast to Rb1 phosphorylation, very little is known about the dephosphorylation of this protein. Two major phosphatases, namely PP1 and PP2A, were described for Rb1 dephosphorylation [183]. Although both phosphatases can dephosphorylate Rb1 in vitro and in vivo, the functional relevance and upstream activation of them over Rb1 differs. Several reports identified PP1 as a major Rb1 phosphatase during cell cycle progression and mitotic exit [184,185]. Moreover, constitutively active PP1 arrests cells in the G1 phase and this effect is dependent on Rb1 [186]. Rb1 binds to PP1 through its C-terminal part and the binding is not dependent on the Rb1 phosphorylation level because both hypophosphorylated and hyperphosphorylated Rb1 can interact with this phosphatase. However, the phosphorylation of certain residues, such as S249, T373, S788, S795, T811, T821 and T826, can negatively affect the interaction of PP1 with Rb1 [187,188]. Moreover, similarly to the ordered fashion of Rb1 phosphorylation, dephosphorylation by PP1 is also sequential and temporally regulated [189,190]. This spatiotemporal regulation of Rb1 dephosphorylation can arise from the different affinities of PP1 isoforms or the presence or absence of different PP1 regulatory subunits [191,192]. For example, two regulatory PP1 subunits, MYPT and SPN, play a role in a tumor suppressor that is mediated by stimulating Rb1 dephosphorylation [191,193].

A second phosphatase involved in mediating Rb1 dephosphorylation is PP2A. Two complementary mechanisms of action were proposed for this phosphatase role. On the one hand, PP2A dephosphorylates CDKs, thereby resulting in lower phosphorylation activity towards Rb1. This mechanism is present mainly in normally cycling, unchallenged cells [194]. On the other hand, it was shown that the major role of PP2A in the regulation of Rb1 phosphorylation is through its direct dephosphorylation as a response to cellular stress, such as DNA damage, ionizing radiation and oxidative stress [195,196,197]. However, this dephosphorylation seems to be a swift event affecting global Rb1 phosphorylation and, unlike PP1 it is not sequential or site-specific.

### 4.2. Acetylation, Methylation and Sumoylation

In addition to phosphorylation, various other post-translational modifications, namely acetylation, methylation and sumoylation, modulate Rb1 function. Several high throughput proteomic studies identified six acetylated lysine residues on the Rb1 protein [198,199]. However, acetylation of only two of the lysines (K873/874) was studied and functionally annotated. These residues are located in the C-terminal pocket domain of Rb1, in the vicinity of major CDK-directed phosphorylation sites, thus their acetylation was expected to alter Rb1 function. Indeed, acetylation of K873/874 by p300/CBP or by P/CAF impairs Rb1 phosphorylation by CDKs and, also, increases Rb1 binding to the ubiquitin ligase MDM2 (Mouse double minute 2 homolog) [199]. This acetylation was present mostly in the terminally differentiated cells, which implies that acetylation is a more stable way to achieve the cell cycle arrest necessary for cellular differentiation [148]. Moreover, it was shown that DNA damage stimulates the acetylation of Rb1, which would result in disassociation of the Rb1–E2F1 complex [147]. However, additional layers of regulation are needed as the interaction of Rb1 with E2F1 on apoptotic genes is important for survival upon DNA damage [20,200,201].

Similar to acetylation, lysine and arginine methylation also affect the functionality of Rb1. The most studied site of methylation is K810. This conserved residue lies next to an important CDK phosphorylation site, S807/811, thus implying functional interaction between these post-translational modifications. K810 is methylated by the methyltransferase Set7/9 and this methylation prevents CDK binding and subsequent Rb1 phosphorylation. Moreover, this methylation is stimulated after DNA damage and it appears to play a crucial role in cell cycle arrest upon DNA damage, by two mechanisms. The first would be the promotion of canonical binding to E2F1, followed by recruitment of HP1 and repression of cell cycle-promoting genes [141]. The second mechanism would be the recruitment of 53BP1 (p53 binding protein 1), which is stimulated directly by Rb1 methylation, thus triggering DNA damage repair [202]. Interestingly, on the other hand, arginine methylation stimulated by PRMT4 on Rb1 R775, R787, and R798 has opposite effects. Arginine methylation of Rb1 is required for full phosphorylation of Rb1 by CDKs, disassociation from E2F1, and efficient cell cycle progression [141].

Finally, Rb1 can be also subjected to SUMOylation. This post-translational modification is blocked by interaction with small viral proteins; however, the significance of this modification has not been addressed in depth [134,135].

## 5. Concluding Remarks

Rb1 is a multifunctional protein that integrates signals from multiple upstream events and plays important roles beyond the regulation of the cell cycle. Rb1 is extensively modified by phosphorylation, acetylation and methylation. Although the interplay between the qualitative and quantitative extent of these modifications is only now starting to be uncovered, a highly complex regulation of the Rb1 function is emerging. It is becoming clear that the modification of the same residue can result in different outcomes depending on other modifications present, and that there are pools or subpopulations of Rb1 molecules that are differentially modified and play different roles. It could be postulated that the downstream functional output of Rb1 is regulated by a “post-translational modification code”. This mechanism of action can be illustrated by the differential output of Rb1–E2F1 interaction in cycling cells and cells undergoing DNA damage. Blocking Rb1 hyperphosphorylation, which would lead to the increased interaction of Rb1 and E2F1, is a favorable outcome on the promoters of cell cycle genes, but it is unfavorable at the pro-apoptotic gene promoters and could lead to insensitivity to DNA damage. Thus, deciphering the “post-translational modification code” of Rb1 might prove crucial to decouple these two signaling events. Moreover, it was shown that some modifications are dominant over others and that they determine the overall output independently of the other ones present.

The importance of the Rb1–E2F signaling axis in cancer is well recognized, however, the only clinically approved drugs targeting the pathway are CDK inhibitors. The efficacy of these inhibitors is dependent on the Rb1 status, thus they cannot be used for patients with non-functional Rb1 protein (deletion, promoter hypermethylation, inactivating mutations) [93]. On the other hand, Rb1 deficiency offers therapeutic opportunities associated with higher sensitivity of such tumors to DNA damaging therapy or microtubule drugs. For example, breast cancer patients with Rb loss exhibited better clinical responses to radiotherapy and systemic chemotherapy [170,172,203,204].

In summary, the crucial role of Rb1 in cell cycle regulation calls for a deeper understanding of its modifications, which in turn would contribute to the design of targeted compounds of therapeutic interest.

## Figures and Tables

**Figure 1 cancers-14-01265-f001:**
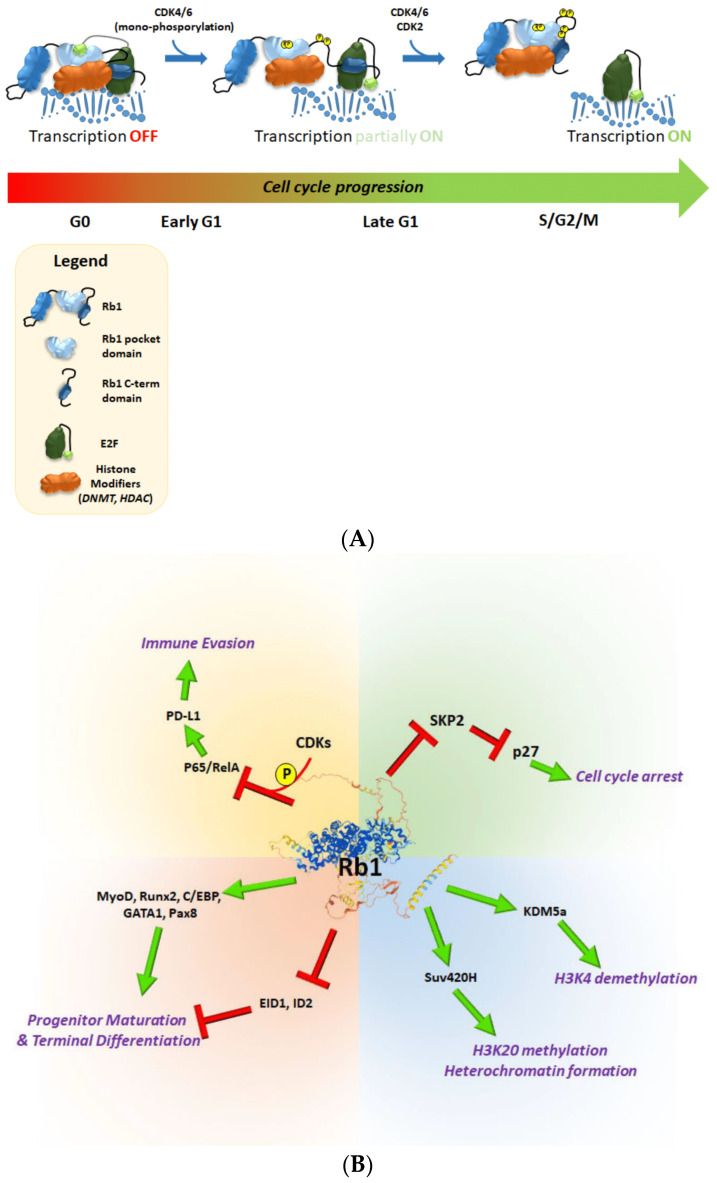
Rb1 regulation during cell cycle progression and downstream signaling (**A**) E2F regulation by Rb1.In the absence of phosphorylation, Rb1 binds to E2F transcription factor through the Rb1 pocket and C-terminal domains and inhibits its transcriptional activity. Moreover, Rb1 recruits histone modification enzymes such as DNMT1, HDACs or Suv39H to block transcription from E2F target promoters by stimulating a repressive chromatin state. Upon entry into cell cycle, Rb1 is phosphorylated by CDK4/6-Cyclin D complexes, which allows partial transcription of E2F-stimulated genes. In late G1 and S phases, Rb1 is hyperphosphorylated by CDK4/6-Cycline D and CDK2-Cycline E complexes, which leads to the disassociation of Rb1 from E2F, full stimulation of E2F target genes, and cell cycle progression. (**B**) E2F-independent Rb1 function: Beyond binding and inhibiting E2F, Rb1 was implicated in the regulation of several signaling pathways: (i) it binds and inhibits p65 transcription factor, which leads to impairment of PD-L1 expression and immune evasion; (ii) it stabilizes cyclin-dependent kinase inhibitor p27 by sequestering SKP2 and targeting it for degradation; (iii) it stimulates terminal differentiation of multiple cell lineages by activating differentiation transcription factors (TFs) such as MyoD, Runx2, GATA1, Pax8 or impairing the function of differentiation inhibitory TFs such as EID1 or ID2; and (iv) it regulates heterochromatinization by recruiting and stimulating the activity of KDM5a demethylase and SUV420H methylase.

**Figure 2 cancers-14-01265-f002:**
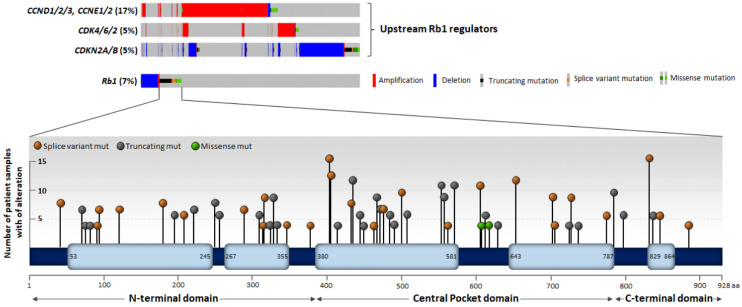
Rb1 axis gene alteration in cancer. The major constituents of the Rb1 signaling axis (Cyclins, CDKs, CKIs, Rb1) are significantly altered in human cancers. Major upstream alterations are *Cyclin* and *CDK* gene amplification and *CDKN2A/B* deletion, which results in hyperactivation of CDK and functional inactivation of Rb1. Moreover, *Rb1* is altered in around 7% of tumors across different types of cancer. Roughly half of the alterations are *Rb1* deletion and the other half are point mutations. The latter is distributed throughout the whole gene with no apparent hotspots. Of note, the vast majority of these mutations are truncations or splice variant mutations, but not substitutions, which would result in expression of an Rb1 protein lacking certain regions.

**Figure 3 cancers-14-01265-f003:**
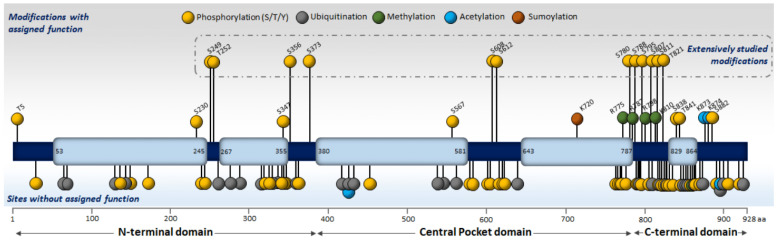
Rb1 post-translational modification. To date, 105 residues on Rb1 protein have been shown to be post-translationally modified. Roughly a quarter of these modifications (upper part) have assigned functions, and 12 of them (all phosphorylations) have been extensively studied (S249, T252, S356, S373, S608, S612, S780, S788, S795, S807, S811, T821).

**Table 1 cancers-14-01265-t001:** Rb1 post-translational modifications and their role in regulating cellular signaling.

Site	Modification	Modifier	Molecular Function	Outcome	Reference
T5	Phosphorylation	N.A.	N.A.	Apoptosis inhibition	[124]
S230	Phosphorylation	N.A.	N.A.	Apoptosis inhibition	[124]
S249	Phosphorylation	p38α	Increase affinity to E2F1	Cell cycle inhibition	[125]
		CDK1/2	Reduce affinity to HDAC5 and EID1	Transcription regulation	[31] [126]
T252	Phosphorylation	p38α	Increase affinity to E2F1	Cell cycle inhibition	[125]
		CDK1/2	Reduce affinity to HDAC5 and EID1	Transcription regulation	[31] [126]
S347	Phosphorylation	N/A	Promote caspase cleavage	Increased Rb proteolysis during apoptosis	[127]
T356	Phosphorylation	CDK2	Reduce affinity to E2F1	Cell cycle entry	[124,128]
T373	Phosphorylation	CDKs	Reduce affinity to E2F1	Cell cycle entry	[128]
S567	Phosphorylation	p38α	Increase affinity to HDM2	Rb degradation and apoptosis	[129,130]
S608	Phosphorylation	CDK2	Reduce affinity to E2F1	Cell cycle entry	[128,131,132]
S612	Phosphorylation	CDK2	Reduce affinity to E2F1	Cell cycle entry	[128]
		Chk1/ Chk2	Increase affinity to E2F1	Cell survival upon DNA damage	[133]
K720	Sumoylation	N/A	Reduce affinity to E2F1	Cell cycle entry	[134,135]
R775	Methylation	PRMT4	Reduce affinity to E2F1	Cell cycle entry	[136]
S780	Phosphorylation	CDK4	Reduce affinity to E2F1	Cell cycle entry	[137]
		Aurora B	Increase affinity to E2F1	Prevents endoreduplication	[138]
		TG2	Reduce affinity to E2F1	Cell cycle entry	[139]
R787	Methylation	PRMT4	Reduce affinity to E2F1	Cell cycle entry	[136]
S788	Phosphorylation	CDKs	Reduce affinity to E2F1	Cell cycle entry	[23]
S795	Phosphorylation	CDK4	Reduce affinity to E2F1	Cell cycle entry	[23]
R798	Methylation	PRMT4	Reduce affinity to E2F1	Cell cycle entry	[136]
Y805	Phosphorylation	Abl tyrosine kinase	N.A.	Necessary for survival of Abl-dependent tumor cells	[140]
S807	Phosphorylation	CDKs	Reduce affinity to E2F1	Cell cycle entry	[131]
K810	Methylation	Set7/9 Smyd2	Inhibits Cdk-directed phosphorylation	Cell cycle arrest	[141,142]
S811	Phosphorylation	CDKs	Reduce affinity to E2F1	Cell cycle entry	[131,143]
T821	Phosphorylation	CDK2	Reduce affinity to E2F1	Cell cycle entry	[23]
			Reduce affinity to HDAC5		[31]
T826	Phosphorylation	CDKs	Reduce affinity to E2F1	Cell cycle entry	[23]
S838	Phosphorylation	p38α	Disrupts condensin II interaction with chromatin	Chromatin decondensation	[144,145]
T841	Phosphorylation	p38α	Disrupts condensin II interaction with chromatin	Chromatin decondensation	[144,146]
K873	Acetylation	N/A	Increase affinity to MDM2 Reduce affinity to E2F1	Cell cycle exit and cell differentiation	[147,148]
K874	Acetylation	N/A	Increase affinity to MDM2	Cell cycle exit and cell differentiation	[147,148]
S882	Phosphorylation	N.A.	Promote caspase cleavage	Increased Rb proteolysis	[127,147]
**Rb modification without assigned function (identified through HTP proteomics)**					
S350	Phosphorylation				[145]
T353	Phosphorylation				[149,150,151]
S624	Phosphorylation				[145]
T774	Phosphorylation				[152]
T778	Phosphorylation				[145,152]
Y790	Phosphorylation				[152,153]
Y813	Phosphorylation				[153]
S816	Phosphorylation				[154,155,156]
T823	Phosphorylation				[146,156,157]
855	Phosphorylation				[145,158,159]

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
