# Peer review of "Understanding Retinoblastoma Post-Translational Regulation for the Design of Targeted Cancer Therapies"

_cancers, 2022, doi:10.3390/cancers14051265_

Round 1

Reviewer 1 Report

This manuscript is a nice review of the RB tumor suppressor and its regulation by post-translational modifications.  The authors summarize the function of RB in regulating cell cycle progression and its regulation by cyclin D-CDK4/6 and cyclin E--CDK2 kinases. One criticism however, is the schematic model in Figure 1A suggests that CDK4/6 phosphorylates multiple residues on the same RB molecule. As discussed later in the this review, cyclin D-CDK4/6 complexes only mono-phosphorylate RB on one of several possible residues. The authors do correctly show in their model that when CDK4/6 mono-phosphorylates RB, it remains bound to E2F transcription factors. Only when CDK2 hyper-phosphorylates RB does RB release E2F. Another general criticism is that the authors focus too much on E2F1 when, as they briefly discuss, E2F1 is part of a family of transcription factors and RB and RB-related proteins regulate most of E2F family members. In particular, most target genes are regulated by multiple E2F family members dependent on the context. Specific issues are:

  1. For the reason mentioned above, the authors should consider using "E2F-target genes" instead of "E2F1-target genes" throughout the manuscript. Also, in Figure 1A consider using E2F instead E2F1 in schematic and legend.
  2. Line 42: Rbl1 (p130) should be Rbl1(p107) and Rbl2(p107) should be Rbl2(p130). The definitions are correct in the abstract.
  3.  Line 164: reference 50 is incorrect. That paper does not refer to E2F but refers to a different transcription factor, p120E4F, which is unrelated to the E2F family. Perhaps that reference can be used in the next section when discussing other transcription factors regulated by RB. 
  4. Line 187, needs space between "on the".
  5. Consider including RB's function in silencing repetitive elements (with EZH2) in the Non-canonical functions of Rb1 section.
  6. The authors mention that ubiquitination is the second most common RB modification but they do not discuss it later. Does ubiquitination only regulate RB stability or does it have other functions?
  7. Line 554: There seems to be an issue with a Carr et al. reference. Should reference 203 go here? If so, please confirm reference 203 is correct on line 557.

Reviewer 2 Report

The review by Janostiak et al is a well written and comprehensive review on post-translational regulation of Rb1. The references are current and well suited to the topic. The tables and figures give a general yet detailed overview of Rb1 post-translational modifications and their impact on Rb1 binding partners and downstream signaling. The discussion of potential avenues for the design of novel cancer therapies based on modulating function was scattered throughout the manuscript and could use more detail since this concept is highlighted in the title and abstract. For example are there studies that examine the efficacy of targeting Rb1 inactivation and DNA damage agents (referring to text lines 299-302 and 320-322)? It would be helpful to have more examples that highlight the current clinical research to support the concept of Rb1-targeted therapies.

Minor revisions:

-line 107 contains a typo

-line 218 has a duplicate reference
